# Association Between Reconstruction Technique and Clinical Outcomes in Advanced Gastric Cancer Patients Undergoing Proximal Gastrectomy

**DOI:** 10.3390/cancers16244282

**Published:** 2024-12-23

**Authors:** Katarzyna Sędłak, Karol Rawicz-Pruszyński, Zuzanna Pelc, Radosław Mlak, Katarzyna Gęca, Magdalena Skórzewska, Krzysztof Zinkiewicz, Katarzyna Chawrylak, Wojciech P. Polkowski

**Affiliations:** 1Department of Surgical Oncology, Medical University of Lublin, 20-080 Lublin, Poland; krpruszynski@gmail.com (K.R.-P.); zuzanna.pelc@umlub.pl (Z.P.); kasiaa.geca@gmail.com (K.G.); magdalena.skorzewska@umlub.pl (M.S.); krzysztof.zinkiewicz@umlub.pl (K.Z.); katchawrylak@gmail.com (K.C.); wojciech.polkowski@umlub.pl (W.P.P.); 2Department of Laboratory Diagnostics, Medical University of Lublin, 20-093 Lublin, Poland; radoslaw.mlak@umlub.pl

**Keywords:** gastric cancer, proximal gastric cancer, double-tract reconstruction, esophagogastrostomy

## Abstract

The incidence and localization of gastric cancer (GC) are shifting toward the proximal stomach, leading to an increased use of proximal gastrectomy (PG) as a surgical option for upper-third GC. This study evaluated textbook outcomes (TOs)—a composite measure of surgical quality—after two reconstruction methods following PG: double-tract reconstruction (DTR) and posterior esophagogastrostomy with partial neo-fundoplication (EGF). A total of 60 patients (30 in each group) from a prospectively maintained database were analyzed. Patients undergoing DTR were 5.5 times more likely to achieve TO compared to those receiving EGF (OR = 5.67; *p* = 0.0266). However, there were no significant differences in overall survival between the two groups. These findings suggest that DTR may be a more favorable reconstruction technique for achieving optimal short-term surgical outcomes. Further randomized controlled trials are needed to confirm these results and guide the selection of the most appropriate reconstruction method after PG.

## 1. Introduction

Gastric cancer (GC) remains a major challenge to public health on a global scale. In 2018, there were nearly 1,000,000 new cases of GC registered worldwide and nearly 800,000 deaths [1]. Due to the COVID-19 pandemic, malignancies have become an even greater threat due to reduced access to health care and a significant deterioration in the health of the general population [2,3]. The anatomical classification of GC is determined by the location of the primary tumor [4]. The rates of distal GC have been steadily declining over the last 50 years in most populations [1], whereas the incidence of proximal GC localization has been increasing, particularly in high-income countries [5]. Risk factors for proximal GC, such as obesity, higher socioeconomic status, and higher education, are more common in Western populations [5,6]. Early infiltration into the esophagus and regional lymphatic dissemination reflect the aggressive behavior of proximal GC [4]. Proximal GC is more common in Europe, Latin America, and North Africa than in Asian–Pacific countries, where the antral location is predominant [7]. National Comprehensive Cancer Network (NCCN) guidelines propose radical gastrectomy with complete lymphadenectomy for patients with stage II to III GC and describe perioperative chemotherapy as a category 1 recommendation for localized GC [8]. The ESMO Clinical Practice Guidelines for stage IB–III GC recommend radical gastrectomy with perioperative chemotherapy [9]. Proximal gastrectomy (PG) has been advocated as an alternative operation for some junctional carcinomas that may be performed as a function-preserving procedure with various new reconstruction techniques [10]. However, in Europe, PG is rarely performed, possibly due to the concern of postoperative reflux, anastomotic stenosis, and meal-related distress [11]. A recent study from the US based on the National Cancer Database (NCDB) confirmed that PG was applied in 68.8% of patients with proximal GC [12].

Constant improvement in minimally invasive techniques for GC patients, followed by various modifications of anti-reflux procedures, may be the reason for the recent popularity of PG. Despite the numerous studies comparing total gastrectomy (TG) and PG for proximal GC, none of them provide a definite answer regarding which method should be implemented [13,14,15]. Japanese Gastric Cancer Association guidelines and Korean Gastric Cancer Association guidelines are the only national guidelines favoring PG in early GC [16,17]. Following PG, several reconstruction methods can be applied, including double-tract reconstruction (DTR), esophagogastrostomy, and jejunal interposition [16]. DTR was first described by Aikou et al. in Japan as a reasonable and simple method which can maintain physiological pancreatocibal synchronism [18] (Figure 1). DTR effectively prevents the reflux symptoms frequently occurring after PG with esophagogastrostomy in early GC [15].

Recently, we introduced a novel technique of posterior esophagogastrostomy and partial neo-fundoplication (EGF) to prevent severe postoperative reflux symptoms in advanced GC patients [19] (Figure 2). In the study, nearly half of the patients were not affected by any reflux symptoms, whereas the others were effectively treated with anti-reflux medication. Nevertheless, the evidence for PG in advanced cancer of the proximal stomach is very limited [19,20].

The textbook outcome (TO) has been proposed to optimize and compare treatment outcomes. It was introduced by the Dutch Upper Gastrointestinal Cancer Audit (DUCA) in esophagogastric cancer surgery as a composite measure consisting of standard surgical indicators [21]. Since TO consists of universal surgical metrics, it can be freely used to compare different types of surgical procedures. In population-based studies, overall survival (OS) was significantly better in patients who achieved TO compared with those who did not [22,23]. The DUCA definition of TO has subsequently been used by other national registries [24].

The aim of the present study was to compare the TO of two reconstruction methods following PG: DTR and EGF.

## 2. Materials and Methods

After receiving institutional review board approval (Bioethics Committee of the Medical University of Lublin, Ethics Code: KE-0254/297/2018), we collected data from a prospectively maintained database of 378 consecutive patients operated upon for GC between May 2010 and May 2024 in the Department of Surgical Oncology of the Medical University of Lublin (Poland). All procedures were performed in accordance with the ethical standards of the institutional and national research committee and with the 1964 Helsinki Declaration and its later amendments or comparable ethical standards. Informed consent was received from all the participants. Patients were scheduled for surgery after a decision had been made by a multidisciplinary team. The inclusion criteria were histologically confirmed, potentially curable, locally advanced primary proximal gastric adenocarcinoma patients who had undergone PG with DTR or EGF. The exclusion criterion was an incomplete histopathology report. Out of 112 patients diagnosed with proximal GC, 60 patients underwent PG and were eligible for analysis. According to the primary aim of this study, we created two study groups: patients in whom DTR was used as a reconstruction method after PG (DTR group) and patients in whom the EGF method was applied for this purpose (EGF group).

### 2.1. Surgical Technique of Proximal Gastric Resection

PG was performed through a midline abdominal incision with wide anterior splitting of the diaphragmatic hiatus and included the resection of the esophagogastric junction and proximal stomach together with two-thirds of the lesser curvature. Preservation of the vagal nerves was not attempted. A suprapancreatic D1plus lymphadenectomy was applied and comprised en bloc removal of all lymphatic tissue along the cardia and fundus and of the lymph nodes (LNs) along the left gastric artery together with LNs along the lesser curvature by cutting the left gastric artery close to its origin and along the common hepatic and splenic artery toward the celiac axis. Thus, during PG, at least the following lymph node stations were removed: 1, 2, 3a, 4sa, 4sb, 7, 8a, 9, 11p [19].

### 2.2. Double-Tract Reconstruction

For DTR, the jejunum distal to the Treitz ligament was transected. The distal jejunal limb was transposed into esophageal hiatus through the left transverse mesocolon in order to achieve esophago-jejunostomy [25]. An end-to-side esophago-jejunostomy was performed with a 25 mm circular stapler, and the jejunal stump was closed with a linear stapler. The integrity of the anastomosis was tested as described in our previous work [26]. Next, an end-to-side gastro-jejunostomy (using only the lower half of the gastric remnant transection line) was performed 15 cm below the esophago-jejunostomy. Finally, an end-to-side jejuno-jejunostomy was performed 15–20 cm below the gastro-jejunostomy (Figure 1).

### 2.3. Reconstruction with Posterior Esophagogastrostomy and Partial Neo-Fundoplication 

Once the creation of a short gastric tube was completed, the distal stump of the esophagus was mobilized through the hiatus. The construction of an esophagogastric anastomosis was performed with a curved intraluminal circular stapler which was introduced through a minimal incision at the most proximal end of the gastric tube and transfixed through the posterior wall of the gastric remnant. After checking the completeness of the stapler donuts, anastomotic integrity was tested intraoperatively by the methylene blue test, as described before [27]. A modified 270-degree fundoplication was then created as an anti-reflux procedure. The technique has been described in detail in our previous work [19] (Figure 1).

### 2.4. Choice of Reconstruction Method

PG with EGF was used in some high-risk patients with serious comorbidities or age > 75 years in whom PG was considered less risky than TG. In patients requiring the resection of the distal part of the esophagus or in patients whose gastric remnant was less than 50% (antrum-preserving), we performed DTR. The final decision to perform either EGF or DTR was made after laparotomy by an experienced surgeon with verification of resectability including downstaging by neoadjuvant chemotherapy, as well as intraoperative molecular cytology of peritoneal fluid performed for a second time after the one performed during staging laparoscopy. Large tumors (with no regression) for which total gastrectomy (or esophagectomy) would be required to guarantee a negative distal (proximal) resection margin were not suitable for this technique. 

### 2.5. Study Group

Baseline characteristics and a comparison of the study groups in terms of demographic and clinical variables are presented in Table 1.

Sixty patients have been included, mostly with intestinal type advanced GC, of whom 73.3% received neoadjuvant chemotherapy. Despite this, 50% of the patients have been found to have LN metastases on final pathological report of the resection specimen.

### 2.6. Perioperative Chemotherapy

Patients were scheduled for treatment based on a combination of platinum and fluoropyrimidine derivatives. The preferred regimen was FLOT-4 consisting of docetaxel at 50 mg/m^2^ on day 1, oxaliplatin at 85 mg/m^2^ on day 1, leucovorin at 200 mg/m^2^ on day 1, and 5-fluorouracil at 2600 mg/m^2^ on day 1 of the cycle, repeated every 14 days [28]. Given the inclusion period and comorbidities excluding FLOT-4 administration, some patients received an EOX/ECF regimen (50 mg/m^2^ epirubicin and 130 mg/m^2^ oxaliplatin on day 1, with 625 mg/m^2^ capecitabine administered twice daily on days 1–21, repeated every three weeks). After 4–5-week intervals, patients were referred for surgical treatment.

### 2.7. Definition of Textbook Outcome

The definition of TO, established by DUCA, consists of 10 parameters: (1) radical resection according to the surgeons’ assessment at the end of the operation, (2) no intraoperative complications, (3) negative resection margins (R0), (4) at least 15 LNs retrieved and examined (surgical compliance), (5) no severe postoperative complications, (6) no re-interventions, (7) no readmission to the intensive care unit (ICU), (8) no prolonged hospital stay (>21 days), (9) no postoperative mortality, and (10) no hospital readmission [21]. We used the modified definition from our previous work on textbook oncological outcomes, which includes perioperative chemotherapy compliance and has been extensively discussed among all other TO definitions [29,30].

### 2.8. Morbidity Evaluation

The Clavien–Dindo classification [31,32] and Comprehensive Complication Index (CCI) [33] scoring systems were used to assess the severity of postoperative complications. Complications were prospectively recorded and classified according to the list of the Gastrectomy Complications Consensus Group [34]. Severe postoperative complications were defined as those with a CCI score above 30 or a Clavien–Dindo classification above grade II.

### 2.9. Statistical Analysis

MedCalc v.15.8 (MedCalc Software, Ostend, Belgium) was used for the statistical analysis of the data. Most continuous variables had non-normal data distribution (assessed by the D`Agostino–Pearson test). The only exceptions were age and ICU hospitalization time. The data were presented as means with standard deviations (SDs) (normally distributed data) or medians (non-normally distributed data) and a minimum and maximum range. When groups were compared in terms of continuous variables, the *t*-test was used for normally distributed data, whereas the Mann–Whitney U test was used for non-normally distributed data. Dichotomized or categorical variables were expressed as numbers and percentages. OS was defined as the period from the date of surgery to the date of the patient’s death or last follow-up. At the time of the analysis, the median follow-up was 17 months for DTR (range: 0.1–34 months) and 42 months for EGF (range: 1–175 months). In the entire study group, approximately 65% of patients had died or were followed for at least two years. Median OS was 31 months. In the univariable analysis of the chance for TO achievement (overall and individual features), an odds ratio (OR) with a corresponding 95% confidence interval (CI) was calculated. Multivariable analysis of the chance for TO (overall and individual features) achievement was performed using logistic regression. Of all categorized variables in Table 1, only anatomical location and TRG were found to be significantly associated with achieving TO. The backward elimination method showed that, apart from the reconstruction method, only TRG should be used to correct the results in multivariate analysis. The Kaplan–Meier estimation method was used for the generation of survival curves. The log-rank test was used to calculate the hazard ratio (HR) and 95% CI in univariable (OS) analysis. Multivariable analysis for OS was performed using Cox proportional hazards. Of all categorized variables in Table 1, only pT and TRG were found to be significantly associated with OS. The backward elimination method showed that only pT should be used to correct the results in multivariate analysis. In all analyses, two-tailed *p*-tests were used. Results with a *p*-value below 0.05 were considered to be statistically significant. 

## 3. Results

### 3.1. Baseline Characteristics and Comparison of the Study Groups

Study groups were balanced in terms of basic demographic and clinical variables: sex, age, Lauren`s histological type, pT, pN, pM, metastatic LNs, lymph node ratio (LNR), grading, tumor regression grade (TRG), anatomical localization, hospitalization time, ICU hospitalization, and ICU hospitalization time. The number of LNs retrieved was significantly higher in patients in whom DTR was performed (median: 27 vs. 18 LNs; *p* = 0.0006). Also, CCI was significantly higher in the DTR than in the EGF group (median: 20.9 vs. 0; *p* = 0.0438). Moreover, preoperative chemotherapy was significantly more commonly used in the DTR than in the EGF group (90% vs. 56.7%; *p* = 0.0086) (Table 1).

### 3.2. Textbook Outcome Achievement

The comparison of TO achievement in the DTR and EGF groups and an analysis of the chances of achieving TO depending on the reconstruction method are presented in Table 2.

### 3.3. Univariable Analysis

Patients undergoing DTR had a 4-fold higher chance of achieving overall TO compared to patients subjected to EGF (OR = 4.12; *p* = 0.0108). Out of ten TO features, only in one (>15 LNs retrieved) was a significant difference found. The use of DTR was associated with a nearly 4-fold higher chance of retrieving a minimum of 15 LNs (OR = 3.76; *p* = 0.0438). 

### 3.4. Multivariable Analysis

Patients undergoing DTR had a 5.5-fold higher chance of achieving overall TO compared to patients subjected to EGF (OR = 5.67; *p* = 0.0266). No significant differences in the chances of achieving any TO component feature depending on the reconstruction method were noted.

No statistically significant differences in OS were noted when both reconstruction methods were compared (Figure 2). 

## 4. Discussion

To the best of our knowledge, this is the first study to compare two reconstruction techniques, i.e., DTR and EGF, in advanced GC patients after PG in a Western population. No significant differences in TO achievement or OS were found. The only significant difference between the reconstruction techniques was in the median number of retrieved LNs. DTR resulted in a nearly doubled LN harvest compared to EGF. A possible explanation for this phenomenon is the greater stomach remnant which is required for EGF reconstruction, comprising stations 3b and 4d. However, the number of metastatic LNs was not significantly different between the compared groups. In patients with cT1/2 and tumors < 4 cm without neoadjuvant chemotherapy, the shortest distance from the pylorus ring to the distal edge of the tumor has been proposed as a predictive factor for perigastric LN metastases in stations 4d, 5, and 6 [35]. The removal of a greater number of LNs during gastrectomy with DTR may indicate that this is an optimal method in oncological terms. 

Given the smaller extent of gastrectomy with EGF, this reconstruction method may be more suitable for older patients with multiple comorbidities, as it potentially reduces the physiological burden of surgery compared to DTR. This assumption aligns with the observation that preoperative chemotherapy was more commonly administered in patients undergoing DTR, suggesting that these patients were likely in better general health and better able to tolerate a more extensive procedure. Furthermore, the smaller extent of surgical resection with EGF may reduce perioperative risks, which is particularly advantageous in frail patients. However, the appropriateness of EGF for such patients needs to be thoroughly assessed in future studies with larger cohorts, standardized patient selection criteria, and prospective designs to avoid bias. Evaluating these techniques in the context of long-term outcomes, quality of life, and nutritional impact will be crucial for refining patient-specific treatment strategies.

There are few studies comparing DTR and EGF in terms of postoperative complications, nutritional outcomes, and body weight loss [20,36,37]. Lu et. al. described that PG with DTR reduces the incidence of reflux esophagitis, anastomotic stenosis, dumping syndrome, and gastric emptying disorder and improves the nutritional status of patients after surgery (either open, laparoscopic, or robotic) [14]. Although we did not investigate long-term effects in our study, further studies on this topic have been planned. Review by Lu et al. on functional digestive tract reconstruction methods after PG described several methods of esophagogastrostomy: tube-like stomach esophagogastrostomy, double-flap technique, and side overlap with fundoplication by Yamashita (SOFY) [14]. However, each of these methods differs to some extent from the method used in our study due to the use of neo-fundoplication. 

Tominaga et al. described the nutritional effects of DTR and EGF following PG. Their study shows the superiority of DTR over EGF in terms of weight loss after surgical treatment [20]. However, we might presume that the use of neo-fundoplication could reduce this imbalance.

Miyauchi et al. found that there was no significant difference between esophagogastrostomy and DTR in terms of operation time, blood loss, and length of postoperative hospital stay [36]. There were also no differences in terms of postoperative complications such as anastomotic leakage, pancreatic fistula, and surgical site infections, which is consistent with the results of our study, which was conducted on a comparable number of patients. Yet, the superiority of DTR over EGF has been demonstrated in the context of short-term nutritional status and preventing postoperative gastroesophageal reflux and anastomotic stenosis. This suggests that studies on a larger number of patients are needed to clearly determine which reconstruction method is most beneficial. 

The findings of our study on the safety and effectiveness of DTR are in agreement with a recent review of the literature, especially in terms of postoperative recovery time, operation time, intraoperative complications, and early complications [38]. In Japan, remnant stomach size and esophageal stump location appear to influence the choice of reconstruction method following PG [39].

Although our study was not designed to detect differences in long-term effects and patient survival between both methods of reconstruction, the results we obtained are promising in the context of both methods. A study evaluating functional effects and quality of life after the application of the two techniques is currently ongoing at our institute. 

This study has several limitations. First, its retrospective design and small sample size introduce selection bias and limit the generalizability of findings. The lack of randomization or propensity score matching may have affected comparability between the groups. Additionally, long-term outcomes, including nutritional status, reflux control, and quality of life, were not evaluated, which limits our understanding of the broader implications of each reconstruction technique.

This study’s focus on a Western population may not be fully applicable to other regions, such as East Asia, where gastric cancer characteristics and surgical practices differ. Moreover, variations in surgeon expertise and institutional protocols were not accounted for, potentially affecting reproducibility in other settings.

Future prospective studies with larger, more diverse cohorts and standardized protocols are needed to confirm these findings and evaluate long-term functional and oncological outcomes comprehensively.

## 5. Conclusions

In patients with proximal GC undergoing PG, TO is more likely to be achieved after DTR than after EGF. Yet, no significant differences in OS were noted when both reconstruction methods were compared. Randomized controlled trials are warranted to indicate the preferred reconstruction technique after PG. 

## Figures and Tables

**Figure 1 cancers-16-04282-f001:**
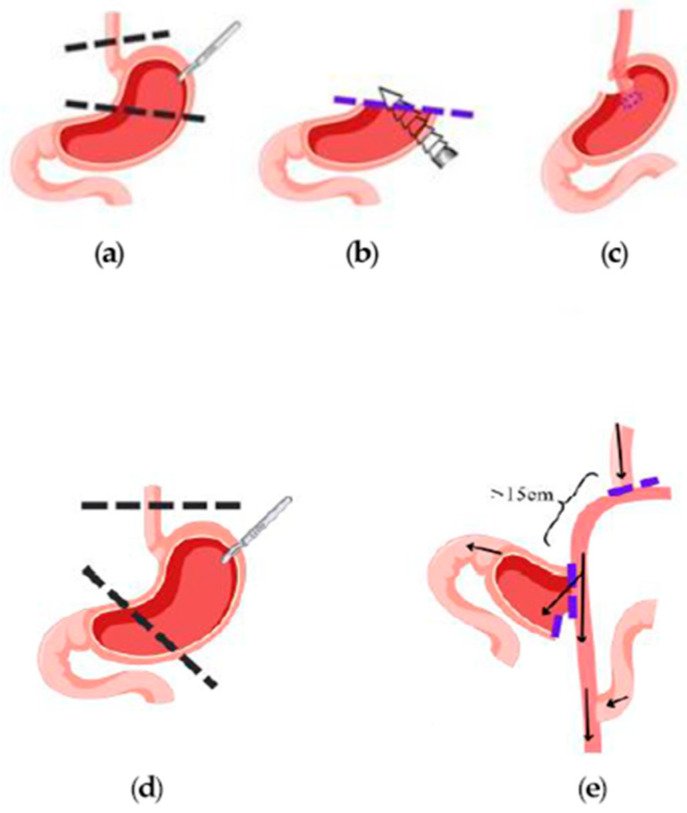
A diagram of the execution of posterior esophagogastrostomy with partial neo-fundoplication and Double-Tract Reconstruction.

**Figure 2 cancers-16-04282-f002:**
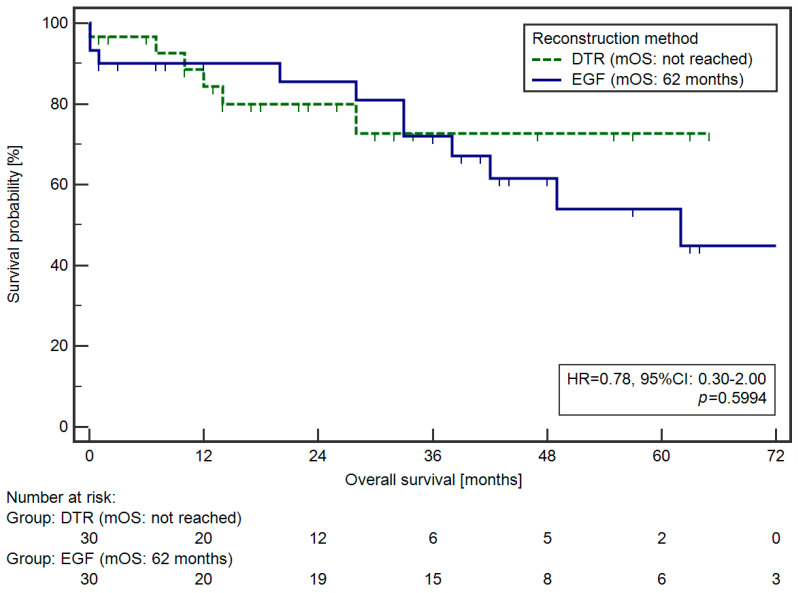
Survival curves of the patients after proximal gastrectomy with EGF and DTR.

**Table 1 cancers-16-04282-t001:** Baseline characteristics and the comparison of the study groups.

Variable	DTR *n* = 30	EGF *n* = 30	*p*
**Sex** Male Female	21 (70%) 9 (30%)	16 (53.3%) 14 (46.7%)	0.2882
**Age [years]** Mean±SD Min–Max	61±8.6 43–79	64.5 ± 12.4 37–84	0.0902
**Age [years]** **<75** **≥75**	27 (90%) 3 (10%)	24 (80%) 6 (20%)	0.4696
**Lauren histological type** Intestinal Mixed Diffuse	18 (60%) 4 (13.3%) 8 (26.7%)	19 (63.3%) 6 (20%) 5 (16.7%)	0.5714
**pT** T0 T1a T1b T2 T3 T4a T4b	4 (13.3%) 0 (0%) 2 (6.7%) 6 (20%) 14 (46.7%) 3 (10%) 1 (3.3%)	3 (10.0%) 2 (6.7%) 2 (6.7%) 2 (6.7%) 14 (46.7%) 5 (16.7%) 2 (6.7%)	0.5469
**pN** N0 N1 N2 N3a N3b	18 (60%) 4 (13.3%) 4 (13.3%) 3 (10%) 1 (3.3%)	14 (46.7%) 9 (30.0%) 5 (16.7%) 1 (3.3%) 1 (3.3%)	0.4727
**pM** M0 M1	29 (96.7%) 1 (3.3%)	29 (96.7%) 1 (3.3%)	0.4720
**Retrieved LNs** Median Min-Max	27 8–51	18 2–37	0.0006 *
**Metastatic LNs** Median Min-Max	0 0–18	0.5 0–14	0.9168
**LNR** Median Min-Max	0 0–0.6	0.01 0–1	0.4362
**Grading** G1 G2 G3	3 (10%) 16 (53.3%) 11 (36.7%)	0 (0%) 20 (66.7%) 10 (33.3%)	0.1745
**Neoadjuvant chemotherapy** Yes No	27 (90%) 3 (10%)	17 (56.7%) 13 (43.3%)	0.0086 *
**Number of neoadjuvant chemotherapy cycles** **Median** **Min-Max**	4 3–8	4 2–5	0.0735
**TRG** 1 2 3 4	4 (16%) 4 (16%) 8 (32%) 9 (36%)	4 (25%) 1 (6.2%) 7 (43.7%) 4 (25%)	0.5921
**Anatomical localization** Esophagogastric junction Upper third of the stomach Middle third of the stomach	4 (13.3%) 26 (86.7%) 0 (0%)	7 (23.3%) 20 (66.7%) 3 (10%)	0.1002
**CCI** Median Min-Max	20.9 0–100	0 0–100	0.0295 *
**Hospitalization time (days)** Median Min-Max	8 5–31	9 5–46	0.6659
**ICU hospitalization** Yes No	2 (6.7%) 28 (93.3%)	2 (6.7%) 28 (93.3%)	0.6048
**ICU hospitalization time (days)** Mean ± SD Min-Max	10 ± 9.9 5–17	5 ± 2.8 3–7	0.6831

*—Statistically significant result. Abbreviations: CCI—Comprehensive Complication Index; DTR—double-tract reconstruction; EGF—posterior esophagogastrostomy with partial neo-fundoplication; ICU—intensive care unit; LNs—lymph nodes; LNR—lymph node ratio; N/a—not applicable; TRG—tumor regression grade.

**Table 2 cancers-16-04282-t002:** Textbook outcome achievement in patients undergoing proximal gastrectomy according to the type of reconstruction.

Textbook Outcome Feature	DTR *n* = 30 (%)	EGF *n* = 30 (%)	Univariable Analysis	Multivariable Analysis
			OR [95%CI] *p*	OR [95%CI] *p*
Overall TO compliance	22 (73.3%)	12 (40%)	4.12 [1.39–12.27] 0.0108 *	5.67 [1.22–26.30] 0.0266 *
Radical resection according to the surgeons’ assessment at the end of the operation	30 (100%)	30 (100%)	1.00 [0.02–52.04] 1.0000	N/a
No intraoperative complications	30 (100%)	30 (100%)	1.00 [0.02–52.04] 1.0000	N/a
Negative resection margins (R0)	28 (93.3%)	26 (86.7%)	2.15 [0.36–12.76] 0.3980	6.00 [0.53–67.28] 0.1462
At least 15 LNs retrieved	26 (86.7%)	19 (63.3%)	3.76 [1.04–13.65] 0.0438 *	1.75 [0.37–8.33] 0.4823
No severe postoperative complication	27 (90%)	27 (90%)	1.00 [0.18–5.40] 1.0000	1.04 [0.15–7.11] 0.9656
No re-interventions	28 (93.3%)	28 (93.3%)	1.00 [0.13–7.60] 1.0000	0.77 [0.06–9.22] 0.8345
No unplanned ICU stay	28 (93.3%)	28 (93.3%)	1.00 [0.13–7.60] 1.0000	0.75 [0.06–9.42] 0.8237
No prolonged hospital stay (21 days)	28 (93.3%)	29 (96.7%)	0.48 [0.04–5.62] 0.5611	0.75 [0.06–9.42] 0.8237
No hospital readmission	30 (100%)	29 (96.7%)	3.10 [0.12–79.23] 0.4936	N/a
No postoperative 30 days mortality	29 (96.7%)	28 (93.3%)	2.07 [0.18–24.15] 0.5611	1.60 [0.09–28.57] 0.7493

*—statistically significant result. Abbreviations: CI—confidence interval; DTR—double-tract reconstruction; EGF—posterior esophagogastrostomy with partial neo-fundoplication; ICU—intensive care unit; LNs—lymph nodes; N/a—not applicable; OR—odds ratio; TO—textbook outcome.

## Data Availability

Data are available from the corresponding author upon request.

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
