# Peer review of "Association Between Reconstruction Technique and Clinical Outcomes in Advanced Gastric Cancer Patients Undergoing Proximal Gastrectomy"

_cancers, 2024, doi:10.3390/cancers16244282_

Round 1

Reviewer 1 Report

Comments and Suggestions for Authors

Dear Authors thank you for submitting your manuscript. I believe it is well written manuscript, and it covers a very interesting topic.

I understand your main goal was to compare techniques to perform a proximal gastrectomy, however I believe it would be interesting to add another group of patients that had a total gastrectomy, in order to compare its outcomes to a more classical approach.

Please consider the following:

Line 41 – “National Comprehensive Cancer Network (NCCN) guidelines propose radical gastrectomy with complete lymphadenectomy for patients with stage II to III GC. ESMO Clinical Practice Guidelines for stage IB–III GC recommend radical gastrectomy with perioperative chemotherapy.” - The way you put this it seems NCCN does not recommend perioperative chemotherapy, which is not the case.

Line 125 – I believe your manuscript would greatly benefit from a schematic picture of the procedures, instead of photographs of the stomach after reconstruction.

Line 128 – This is what I believe is the weakest point of your manuscript. I believe it is not very well explained why you decided to use a specific technique on each patient. The way you describe how you choose the reconstruction method is very confusing.

“PG with EGF was performed in patients with GC located in the upper third of the stomach and when an adequate response to chemotherapy had been achieved” - what about DTR? Did you prefer this method if response to chemotherapy was not achieved?

Line 132 – “In patients requiring resection of the distal part of the oesophagus or in patients whose gastric remnant was less than 50% (antrum preserving), we performed DTR.” – Why you did not prefer EGF in these patients?

Line 136 – “…as well as intraoperative molecular cytology of peritoneal fluid.” Was this not performed during a staging laparoscopy done previously? Do you repeat the cytology, or you only perform it during the resection operation?

Line 146 – You present 2 patients with pM1 disease. What sort of patients were these? Did you know about the M status previously to surgery? Were these palliative surgeries?

Line 146 – pT, pN, pM, Retrieved LNs, Metastatic LNs, LNR, CCI, Hospitalization time, ICU hospitalization and ICU hospitalization time should come in the results section and not in the baseline characteristics of the study groups. If you prefer you can put in this section cTNM but should not put pTNM (as this was only knew after the surgery).

Line 146 – You should state why did you prefer to use DTR in patients who had neoadjuvant chemotherapy (90% vs 56%) as in the “Choice of reconstruction method” section you state:PG with EGF was performed in patients with GC located in the upper third of the stomach and when an adequate response to chemotherapy had been achieved”.

Line 146 – Anatomical localization – when you put “upper third” and “middle third” you should add “of the stomach” to avoid confusion.

Line 146 – Your data shows three patients who had an EGF reconstruction that presented with lesions on the middle third of the stomach. However in the “Choice of reconstruction method” section you state: “In patients requiring resection of the distal part of the oesophagus or in patients whose gastric remnant was less than 50% (antrum preserving), we performed DTR” - this is somehow contradictory as in these 3 patients who presented a more distal tumour you actually performed an EGF reconstruction (and none had a DTR).

Line 153 – “The preferred regimen was FLOT-4 … some patients received an EOX/ECF regimen” – you should state how many cycles were administered.

Line 182 – “At the time of the analysis, the median follow-up was 17 months for DTR (range: 0.1-34 months) and 42 months for EGF (range: 1-175 months).” – why did you have such a difference in the median follow up between techniques? It seems you started performing EGF and then changed your preferred technique to DTR. This brings us to the already mentioned confusing reasons to choose a technique over the other on each particular patient.

Line 230 – The figure 2 notes present 27 patients on EGF group and 30 patients on DTR group, but you mentioned before there were 30 patients on each group. Why are 3 patients missing from the EGF group?

Line 239 – “However, the number of metastatic LNs was not significantly different between compared groups.” - If you mention that the number of harvested lymph nodes is the double with DTR than that with EGF, this statement is somehow biased.

Line 249 – “Thus, it can be suspected that these were patients in better general condition.” – you should have the ECOG status form each patient, then this could eventually be demonstrated and not only “suspected”.

Author Response

Reviewer 1:      Dear Authors thank you for submitting your manuscript. I believe it is well written manuscript, and it covers a very interesting topic.

I understand your main goal was to compare techniques to perform a proximal gastrectomy, however I believe it would be interesting to add another group of patients that had a total gastrectomy, in order to compare its outcomes to a more classical approach.

Please consider the following:

Line 41 – “National Comprehensive Cancer Network (NCCN) guidelines propose radical gastrectomy with complete lymphadenectomy for patients with stage II to III GC. ESMO Clinical Practice Guidelines for stage IB–III GC recommend radical gastrectomy with perioperative chemotherapy.” - The way you put this it seems NCCN does not recommend perioperative chemotherapy, which is not the case.

Line 125 – I believe your manuscript would greatly benefit from a schematic picture of the procedures, instead of photographs of the stomach after reconstruction.

Line 128 – This is what I believe is the weakest point of your manuscript. I believe it is not very well explained why you decided to use a specific technique on each patient. The way you describe how you choose the reconstruction method is very confusing.

“PG with EGF was performed in patients with GC located in the upper third of the stomach and when an adequate response to chemotherapy had been achieved” - what about DTR? Did you prefer this method if response to chemotherapy was not achieved?

Line 132 – “In patients requiring resection of the distal part of the oesophagus or in patients whose gastric remnant was less than 50% (antrum preserving), we performed DTR.” – Why you did not prefer EGF in these patients?

Line 136 – “…as well as intraoperative molecular cytology of peritoneal fluid.” Was this not performed during a staging laparoscopy done previously? Do you repeat the cytology, or you only perform it during the resection operation?

Line 146 – You present 2 patients with pM1 disease. What sort of patients were these? Did you know about the M status previously to surgery? Were these palliative surgeries?

Line 146 – pT, pN, pM, Retrieved LNs, Metastatic LNs, LNR, CCI, Hospitalization time, ICU hospitalization and ICU hospitalization time should come in the results section and not in the baseline characteristics of the study groups. If you prefer you can put in this section cTNM but should not put pTNM (as this was only knew after the surgery).

Line 146 – You should state why did you prefer to use DTR in patients who had neoadjuvant chemotherapy (90% vs 56%) as in the “Choice of reconstruction method” section you state: “PG with EGF was performed in patients with GC located in the upper third of the stomach and when an adequate response to chemotherapy had been achieved”.

Line 146 – Anatomical localization – when you put “upper third” and “middle third” you should add “of the stomach” to avoid confusion.

Line 146 – Your data shows three patients who had an EGF reconstruction that presented with lesions on the middle third of the stomach. However in the “Choice of reconstruction method” section you state: “In patients requiring resection of the distal part of the oesophagus or in patients whose gastric remnant was less than 50% (antrum preserving), we performed DTR” - this is somehow contradictory as in these 3 patients who presented a more distal tumour you actually performed an EGF reconstruction (and none had a DTR).

Line 153 – “The preferred regimen was FLOT-4 … some patients received an EOX/ECF regimen” – you should state how many cycles were administered.

Line 182 – “At the time of the analysis, the median follow-up was 17 months for DTR (range: 0.1-34 months) and 42 months for EGF (range: 1-175 months).” – why did you have such a difference in the median follow up between techniques? It seems you started performing EGF and then changed your preferred technique to DTR. This brings us to the already mentioned confusing reasons to choose a technique over the other on each particular patient.

Line 230 – The figure 2 notes present 27 patients on EGF group and 30 patients on DTR group, but you mentioned before there were 30 patients on each group. Why are 3 patients missing from the EGF group?

Line 239 – “However, the number of metastatic LNs was not significantly different between compared groups.” - If you mention that the number of harvested lymph nodes is the double with DTR than that with EGF, this statement is somehow biased.

Line 249 – “Thus, it can be suspected that these were patients in better general condition.” – you should have the ECOG status form each patient, then this could eventually be demonstrated and not only “suspected”.

Dear Reviewer, thank you for this insightful feedback. We appreciate your acknowledgment of the comprehensive overview of the background and rationale for our study.

  1. The authors would like to thank the reviewer for pointing out this important aspect.

The following change has been introduced into Introduction section.

“National Comprehensive Cancer Network (NCCN) guidelines propose radical gastrectomy with complete lymphadenectomy for patients with stage II to III GC and describe perioperative chemotherapy as category 1 recommendation for localized GC.” (lines 261-262)

  1. The schematic picture with the comparison of the reconstruction techniques has been added.
  2. The authors agree with the Reviewer. If the remnant of the stomach was less than 50% the preferred method of reconstruction was DTR since to perform proper and safe fundoplication a certain size of remnant is required. If the remnant was too small, it was one of the reasons to perform DTR.
  3. The cytology of the peritoneal fluid is always performed during the staging laparoscopy. We repeat this procedure during the gastrectomy.

The following sentence has been modified:

“The final decision to perform either the EGF or the DTR was made after laparotomy by an experienced surgeon with verification of resectability including downstaging by neo-adjuvant chemotherapy, as well as intraoperative molecular cytology of peritoneal fluid being performed a second time after the one performed during staging laparoscopy.” (lines 149-150)

  1. All the patients who were included in the study were classified as the cM0 without any visible metastases. Those two cases described as the pM1 were diagnosed after the histopathological report retrieval and in both cases, they had single peritoneal metastasis.
  2. Table 1. have been moved to Results section.
  3. The use of neoadjuvant chemotherapy had no influence on the choice of the reconstruction method. The disproportion between the use of neoadjuvant chemotherapy (90% vs 56%) in favor of DTR is rather a reflection of the younger age of patients and the smaller number of comorbidities in this group of patients. In these patients, it was easier to decide on DTR reconstruction, which consists of three anastomoses versus EGF which necessitates only one anastomosis.
  4. Table 1. has been corrected.
  5. In these three patients, tumour was indeed located in the middle third of the stomach (corpus), but in the most proximal portion of the corpus and close to the minor curvature. Therefore, PG with adequate distal margin (5 cm) was still possible and the excess of stomach remnant was sufficient to create the floppy partial fundoplication around the esophago-gastric anastomosis (DTR).
  6. The data regarding number of neoadjuvant chemotherapy cycles was added in Table 1.
  7. In some patients, EGF was applied earlier, but over time the two methods were used in parallel.
  8. Figure 2 is a photograph picturing the DTR. In Table 2. we present 30 patients both in DTR and EGF group.
  9. We understand the reviewer's concerns and thank you for this comment. However, in both groups the median number of nodes removed is above the current standard count of 15 lymph nodes. For this reason, the authors suggest that the number of metastatic lymph nodes in both groups may be subject to comparison.
  10. Unfortunately, at this stage of the study, we do not have the ECOG results, but in future studies we will take into account.

We trust that modifications contribute to the overall strength of our study.

Reviewer 2 Report

Comments and Suggestions for Authors

Dear authors,

It was a pleasure to read such a well-executed and thoroughly conducted study. I have several suggestions to improve the presentation of the results:

  1. The paper would benefit from the inclusion of a subsection describing the sample size and sampling techniques. In this subsection, the authors could present the assumptions used for the sample size calculation and provide more details regarding the recruitment of study participants.

  2. In the "Statistical Analysis" subsection, the authors have clearly indicated that most of their continuous variables were non-normally distributed. However, in Table 1, both the mean and median are listed. I suggest presenting either the mean or the median, depending on the results of the normality test for data distribution. Additionally, it is not clear what statistical test was used for the between-group comparisons. I assume it could be either Student's t-test or the Mann-Whitney U-test, but this needs to be clearly specified.

  3. The paper would benefit from an expanded Discussion section, as not all of the study findings have been adequately addressed. Moreover, the authors should provide a more detailed discussion of the study's limitations.

Author Response

Reviewer 2.   It was a pleasure to read such a well-executed and thoroughly conducted study. I have several suggestions to improve the presentation of the results:

The paper would benefit from the inclusion of a subsection describing the sample size and sampling techniques. In this subsection, the authors could present the assumptions used for the sample size calculation and provide more details regarding the recruitment of study participants.

In the "Statistical Analysis" subsection, the authors have clearly indicated that most of their continuous variables were non-normally distributed. However, in Table 1, both the mean and median are listed. I suggest presenting either the mean or the median, depending on the results of the normality test for data distribution. Additionally, it is not clear what statistical test was used for the between-group comparisons. I assume it could be either Student's t-test or the Mann-Whitney U-test, but this needs to be clearly specified.

The paper would benefit from an expanded Discussion section, as not all of the study findings have been adequately addressed. Moreover, the authors should provide a more detailed discussion of the study's limitations.

Dear Reviewer, thank you for this valuable observation and careful insight into our work.

  1. We thank the reviewer for this comment and understand his concerns. However, due to the rarity of proximal gastrectomy in the Western population (approximately 1.5% of all gastrectomies), it remained a challenge to assemble such a group of patients. Therefore, we decided to publish these early data without first using a sample size calculation.
  2. Table 1 was corrected according to the Reviewer`s suggestion (mean and SD or median were presented depending on the data normality assessment results). The explanation on the use of mean and SD or median as well as appropriate tests (t-test or Mann-Whiney U was added in an appropriate paragraph of the Statistical analysis section in the M&M).
  3. Detailed OS results regarding all pT stages separately was added (Figure 2).
  4. The Discussion section have been expanded as follows:

“Given the smaller extent of gastrectomy with EGF, this reconstruction method may be more suitable for older patients with multiple comorbidities, as it potentially reduces the physiological burden of surgery compared to DTR. This assumption aligns with the observation that preoperative chemotherapy was more commonly administered in patients undergoing DTR, suggesting that these patients were likely in better general health and better able to tolerate a more extensive procedure. Furthermore, the smaller extent of surgical resection with EGF may reduce perioperative risks, which is particularly advantageous in frail patients. However, the appropriateness of EGF for such patients needs to be thoroughly assessed in future studies with larger cohorts, standardized patient selection criteria, and prospective designs to avoid bias. Evaluating these techniques in the context of long-term outcomes, quality of life, and nutritional impact will be crucial for refining patient-specific treatment strategies.” (lines 265-276)

“This study has several limitations. First, its retrospective design and small sample size introduce selection bias and limit the generalizability of findings. The lack of ran-domization or propensity score matching may have affected comparability between the groups. Additionally, long-term outcomes, including nutritional status, reflux control, and quality of life, were not evaluated, which limits understanding of the broader im-plications of each reconstruction technique.

The study's focus on a Western population may not be fully applicable to other re-gions, such as East Asia, where gastric cancer characteristics and surgical practices differ. Moreover, variations in surgeon expertise and institutional protocols were not accounted for, potentially affecting reproducibility in other settings.

Future prospective studies with larger, more diverse cohorts and standardized protocols are needed to confirm these findings and evaluate long-term functional and oncological outcomes comprehensively.” (lines 304-316).

Your feedback has been invaluable in refining this aspect of our manuscript, and we appreciate your guidance.

Round 2

Reviewer 1 Report

Comments and Suggestions for Authors

Thank you for reviewing your manuscript, the changes made and the provided  explanations improved it substantially.

I will only suggest you to take into account the following:

Line 278 – You have two figures named “figure 3” (see line 162 please).

Line 278 - When I mentioned in my last review: “The figure 2 notes present 27 patients on EGF group and 30 patients on DTR group, but you mentioned before there were 30 patients on each group. Why are 3 patients missing from the EGF group?”, I meant figure which is now on the line 278 (“Survival curves of the patients after proximal gastrectomy with the EGF and DTR”). You have 27 patients on EGF group and it should be 30. Please address this.

Author Response

I will only suggest you to take into account the following:

Line 278 – You have two figures named “figure 3” (see line 162 please).

Line 278 - When I mentioned in my last review: “The figure 2 notes present 27 patients on EGF group and 30 patients on DTR group, but you mentioned before there were 30 patients on each group. Why are 3 patients missing from the EGF group?”, I meant figure which is now on the line 278 (“Survival curves of the patients after proximal gastrectomy with the EGF and DTR”). You have 27 patients on EGF group and it should be 30. Please address this.

The authors would like to kindly thank the Reviewer for the close insight in our work and for the meticulous review. 

We adressed the comments as follows: 

Line 278 - The omission has been corrected. 

Line 278- The figure is now corrected. 

We believe that after the revision manuscript improved sufficiently.

Reviewer 2 Report

Comments and Suggestions for Authors

Well done!

Author Response

Dear Reviewer, thank you for this insightful feedback. We appreciate your acknowledgment of the comprehensive overview of the background and rationale for our study.